# Invasive Plant Species Biomass—Evaluation of Functional Value

**DOI:** 10.3390/molecules26133814

**Published:** 2021-06-22

**Authors:** Anamarija Peter, Jana Šic Žlabur, Jona Šurić, Sandra Voća, Dubravka Dujmović Purgar, Lato Pezo, Neven Voća

**Affiliations:** 1Department of Agricultural Technology, Storage and Transport, Faculty of Agriculture, University of Zagreb, Svetošimunska cesta 25, 10000 Zagreb, Croatia; apeter@agr.hr (A.P.); jsuric@agr.hr (J.Š.); svoca@agr.hr (S.V.); nvoca@agr.hr (N.V.); 2Department of Agricultural Botany, Faculty of Agriculture, University of Zagreb, Svetošimunska cesta 25, 10000 Zagreb, Croatia; dpurgar@agr.hr; 3Engineering Department, Institute of General and Physical Chemistry, University of Belgrade, Studentski trg 12/V, 11000 Belgrade, Serbia; latopezo@yahoo.co.uk

**Keywords:** phytochemicals, polyphenols, antioxidant capacity, weeds, nonnative species, biowaste

## Abstract

Invasive plant species (IAS), with their numerous negative ecological, health, and economic impacts, represent one of the greatest conservation challenges in the world. Reducing the negative impacts and potentially exploiting the biomass of these plant species can significantly contribute to sustainable management, protect biodiversity, and create a healthy environment. Therefore, the main objective of this study was to evaluate the nutritional potential, phytochemical status, and antioxidant capacity of nine alien invasive plant species: *Abutilon theophrasti*, *Amaranthus retroflexus*, *Ambrosia artemisiifolia*, *Datura stramonium*, *Erigeron annuus*, *Galinsoga ciliata*, *Reynoutria japonica*, *Solidago gigantea*, and *Sorghum halepense*. Multivariate statistical methods such as cluster and PCA were performed to determine possible connections and correlations among selected IAS depending on the phytochemical content. According to the obtained results, *R. japonica* was notable with the highest content of vitamin C (38.46 mg/100 g FW); while *E. annuus* (1365.92 mg GAE/100 g FW) showed the highest values of total polyphenolic compounds. *A. retroflexus* was characterized by the highest content of total chlorophylls (0.26 mg/g) and antioxidant capacity (2221.97 µmol TE/kg). Therefore, it can be concluded that the selected IAS represent nutrient-rich plant material with significant potential for the recovering of bioactive compounds.

## 1. Introduction

Awareness of the need to protect nature and the diversity of life on our planet is key to a healthy and sustainable future. Despite efforts to conserve and sustainably use nature and its sources, the adoption and acceptance of the Convention on Biological Diversity, the Global Biodiversity Review, the IPBES Global Assessment, and the Millennium Ecosystem Assessment, the trend in the protection of ecosystems and biodiversity is still increasingly negative [1,2,3]. Ecosystems’ health and stability are deteriorating, and changes on Earth that took place over millions of years are now happening within a century, even a decade, and there is almost no ecosystem that has not been affected by humans [3,4]. According to some estimates, about 0.25% of the remaining species are lost and disappear each year before they can be identified and studied, while about 34,000 species or even 12.5% of the world’s flora are threatened with extinction [5,6,7]. Preservation of biodiversity is of great importance, particularly in certain areas where specific and native plant species dominate. Human impact is most visible through habitat modification and destruction, ecosystem degradation, pollution, population growth, and, most importantly, transport and migration leading to the introduction and spread of invasive alien species (IAS) [4].

Globally, IAS are considered one of the greatest threats to biodiversity, just after the direct destruction of natural habitats [2,8]. Regarding the species introduced into an area outside their natural habitat as a result of intentional or unintentional human activities, the process of naturalization in a newly colonized area represents one of the “big five” ecological problems of the 21st century [9,10]. IAS do not colonize all habitat types to the same extent but are more often found in habitats with high anthropogenic influence [11]. Very often, they represent the dominant species on ruderal, i.e., marginal and neglected soils, as well as complexes of cultivated land, lawns, pastures, parks, gardens, industrial areas, etc. [12,13]. Globalization, advances in technology, better connections between countries and continents (tourism, transport, trade) have created conditions for their spread to areas where they have never been and would not spread naturally [14]. In recent years, an increasing number of studies have focused on analyzing the characteristics of invasive plants and the characteristics of communities that are vulnerable to invasions and have a significant impact on species, communities, ecosystems, and human well-being [15,16,17,18]. With climate change as one of the most pressing problems of our time, more and more activities are aimed at reducing its impact. Bradley et al. [19] state that climate change may also affect the ability of invasive plant species to survive in certain locations, as well as competition with native species. Furthermore, in addition to ecological (pedological, climatological, botanical, zoological, etc.) effects, impacts on health (allergies, poisoning, and diseases) and economy (agriculture, forestry, tourism, infrastructure, human health, etc.) can increasingly be observed [20].

Agricultural and economic losses due to IAS are also relatively high [21]. There are three different methods for their elimination: mechanical, chemical, and biological. With good planning, the right techniques, and a lot of effort, it is possible to control many invasive species. Unfortunately, all existing control measures have limited success, high implementation costs, possible harmful effects on the environment, or, in most cases, require continuous implementation, i.e., continuous financial costs [21]. Eradication methods are often considered as desirable but at the same time a very difficult and expensive option [22]. With the use of different methods to control IAS, there is a possibility of further spread of side effects as well as further environmental degradation [23]. In the long term, this can be even more detrimental to biodiversity and environmental protection than the invasion itself (risk of further spread through plowing or composting, etc.) [24]. For these reasons, prevention is considered the best option for the management of IAS. Since the amount of biomass they produce increases exponentially from year to year, the costs associated with their management, prevention of spread, removal, and disposal grow proportionally [25].

Instead of being considered a liability, IAS can become an ecosystem service; for example, they can be used to produce various products such as brooms, baskets, other handicrafts, animal feed (chlorophylls, nutrients), medicinal preparations, raw material for firewood, biochar, biogas, etc. [21,26]. These properties of IAS are enabled by certain molecules that help plants to adapt to an altered environment and grow normally; they belong to a category of compounds called specialized metabolites [27,28]. In addition to changes in primary metabolites such as saccharides, nucleic acids, amino acids, and related compounds (betaine), stressed plants can accumulate abundant hydrophilic proteins and polyols in addition to the accumulation of specialized metabolites (phenols, flavonoids, nitrogenous and terpene compounds) [27]. Parkhomenko et al. [29] and others [30,31] mention the possibilities of using IAS in the context of exploiting their functional value (antioxidant and antimicrobial activity of phytonutrients) by extracting certain phytonutrients. Flavonoids, for example, have found wide application in the medical, pharmaceutical, and food industries as pharmacologically active principles; in the food and beverage industry as foaming agents, antioxidants, preservatives, additives, and flavorings; in agriculture mainly as allelochemicals and other important industrial biochemical compounds biosynthesized from primary metabolites. Therefore, it is very important to study these species to evaluate their chemical composition and assess their nutritional potential, which distinguishes them in this context [27,32]. Contrary to the popular belief that IAS and their pollen (e.g., ragweed) are extremely harmful to health, secondary metabolites of IAS could also benefit the human body and health. Sustained adaptation and upgrading of existing IAS control and management measures [21] could potentially utilize them in the form of phytonutrients, specialized metabolites (vitamin C, phenols, flavonoids, nonflavonoids, chlorophylls, carotenoids, respectively), making them drivers of change at ecological, economic, energetic, or even social levels. Weeds or alien invasive plant species can be used as raw materials since large amounts of material are available. Their collection can protect native plants while creating economic uses [33]. Therefore, the main objective of this study was to valorize the nutritional potential, phytochemical status, and antioxidant capacity of nine alien invasive plant species. Pattern recognition techniques (cluster analysis (CA) and principal component analysis (PCA)) were employed in the bioactive compounds dataset (used as descriptors) to characterize and differentiate among the studied samples based on their corresponding botanical origin.

## 2. Results and Discussion

Nine different alien invasive species (IAS) were investigated in this study. Analysis of variance (ANOVA) revealed that all the studied traits differed significantly (*p* ≤ 0.0001) in terms of the specialized metabolites content. The total dry matter content (DM, %) of IAS is presented in Figure 1. DM content of the selected IAS ranged from 14.95 to 30.52%. The highest DM was observed in *S. halepense* (30.52%), followed by *S. giganthea* (29.09%), *A. theophrasti* (26.98%), and *E. annuus* (26.53%). In contrast, the lowest DM content was determined in *G. ciliata* (14.95%). Compared to other literature data [34], which obtained DM values on average of 16.61%, average DM values of analyzed IAS within this research were around 22.46% and are slightly higher. Higher DM values in selected IAS can be explained by the specific pedoclimatic conditions of the investigated area, biotic and abiotic, as well as anthropogenic factors, and genetic characteristics. However, higher DM values are also a first indicator of the nutritional potential of plant materials; therefore, higher DM values take into account a higher amount of chemical compounds (vitamins, minerals, phytochemicals) and thus represent a more nutritionally potent source.

The analysis of the chemical composition of the selected IAS (Figure 2) includes total carbon (C), hydrogen (H), nitrogen (N), sulfur (S), and oxygen (O), as well as ash and protein content in dry IAS biomass. The results of chemical composition differed significantly (*p* ≤ 0.0001) among the IASs studied. The lowest total C (35.49%) was determined in *A. retroflexus*, while the highest (52.39%) in *S. gigantea*. The species *E. annuus* and *D. stramonium* also showed high total C values, on average 50.86%. Total H values in the different IAS species ranged from 4.76% (*A. retroflexus*) to 6.42% (*D. stramonium*); total N content from 0.77% (*E. annuus*) to 7.03% (*R. japonica*); total S content was barely detected in most of the biomass samples and ranged from 0.13% (*S. gigantea*) to 0.32% (*D. stramonium*), while the lowest total O content was determined in *R. japonica* (38.86%), and the highest in *A. retroflexus* (55.62%).

Total protein values in the different IAS species ranged from 4.83% (*E. annuus*) to 43.93% (*R. japonica*), while the ash content of selected IAS ranged from 4.68% (*A. retroflexus*) to 34.37% (*D. stramonium*). Moreover, IAS are often classified as waste and incinerated or inadequately disposed of ecologically without realizing their potential, thus endangering the environment and biodiversity. A higher percentage of ash (mineral residue after combustion) is desirable because the ash can be used as fertilizer or soil improver and plant growth promoter. Higher protein content also contributes to the potential uses of these IAS, while proteins as enzymes also have various functions in the organism, such as protective functions against pests and pathogens in some plants that produce them. The increased carbon content of the plant represents a major enrichment of CO_2_ from the atmosphere, but once the plant is removed from nature, it can be used as feedstock for energy production and biofuels, especially coal and bio-oil or other value-added products, or they can also be returned to the soil as a fertilizer, same as nitrogen. However, care must be taken when choosing the method of conversion and use of IAS, as, for example, energy production from various fuels involves both nitrogen oxide (NOx) and sulfur dioxide (SO_2_) emissions. For this reason, a lower proportion of sulfur is desirable, as certain uses of biomass (e.g., combustion) can produce acid rain due to the formation of SO_2_, which has a negative impact on the environment, ecosystem, and human health, while a higher proportion of hydrogen may indicate the potential of its production from plants in a green and sustainable way without the use of fossil fuels. Considering other literature references [35,36,37], it can be stated that there is a possibility to use IAS based on their chemical and nutritional composition, with the aim of removing them from nature and using them in the extraction of value-added products. Instead of considering IAS as waste, there is a need to explore the possibilities of its use, especially in the field of agriculture as fertilizers or soil improvers in the form of ash, in the production of herbicides, energy, production of coloring agents or medicines, for purposes of phytotherapy, etc.

### 2.1. Bioactive Compounds Content

The way plants interact with other organisms in an environment is very complex. The production and/or accumulation of secondary metabolites can have several functions: self-defense (plant protective mechanisms), sexual attraction, symbiosis, allelopathy, rapid growth, and uncontrollable spread [38]. A higher prevalence of these functional competitive traits allows IAS to survive the colonization phase and spread across a wide range of environments, as support for the process of invasion [33,39].

Vitamin C or ascorbic acid (*AsA*) is an important agent in plant defense mechanism, mainly because of the main function of reactive oxygen species (ROS) detoxification. Due to constant global climate changes and uncertain environmental conditions, plants in their native habitat are often exposed to various stress factors and therefore produce greater amounts of defense phytochemicals, including *AsA* [40]. The *AsA* content determined among the selected IAS biomass (Table 1) ranged from 16 mg/100 g FW (*E. annuus*) to 38.46 mg/100 g FW, recorded in *R. japonica*. From the striking variation between the *AsA* values of IAS, species *R. japonica* had almost 2.5 times higher value than *E. annuus*, while species *A. theophrasti*, *A. retroflexus*, *A. artemisiifolia*, *D. stramonium*, *G. ciliata*, and *S. gigantea* did not differ statistically in *AsA* content, with an average value of 20.3 mg/100 g FW. In general, there is a lack of research on phytochemicals and other specialized metabolites, especially *AsA*, of IAS but authors Sarker and Oba [34] studied *AsA* content in species *A. viridis* and *A. spinosus* and reported values ranging from 44.62 to 107.45 mg/100 g FW, which is in comparison to species studied in this research (*A. retroflexus*) from the same family (Amaranthaceae), a significantly higher value. Based on the stated, it can be concluded that regardless of the species belonging to the same family, their differences in chemical composition are significant, i.e., they also differ significantly in the content of certain specific phytochemicals. Therefore, it is important to note that genetic traits are one of the most important factors affecting the nutrient composition of a plant species and, consequently, its biological and functional properties. Another possible reason for the deviation of the results of *AsA* may be in different environmental and living conditions, i.e., pedoclimatic factors (mainly average temperature, amount of rainfall, etc.) specific for certain areas where IAS are growing. As mentioned, stress conditions (drought stress, pathogens, light exposure) can significantly affect *AsA* synthesis, therefore increasing in content as a direct response of plant organism to defense [41]. Additionally, it is important to emphasize that the main part of the plant responsible for photosynthesis is the leaf; thus, in general, the greater leaf mass of the plant, for example, of *R. japonica*, may potentially have a higher chlorophyll content, resulting in a higher photosynthesis rate. Since glucose is the direct product of photosynthesis and is a precursor for the synthesis of *AsA*, it can be concluded that plant tissues with a higher photosynthesis rate can accumulate higher amounts of *AsA* [42]. Nevertheless, regarding the obtained results, analyzed IAS, as a source of *AsA*, had an average value of 22.37 mg/100 g FW, as compared, for example, with fresh apple (approximately 4.6 mg/100 g FW) [43], spinach (31.6 mg/100 g FW) [44], or peas (30.9 mg/100 g FW) [44], which should not be neglected.

Polyphenols are among the best-known specialized metabolites widely distributed in the plant kingdom and naturally occurring with a wide range of biological activities [45], mainly antioxidant and beneficial effects on health [45,46]. The results of total phenolic content (TPC), total nonflavonoid (TNFC), and flavonoid content (TFC) differed significantly among the IASs studied (Table 1). The lowest TPC (142.66 mg GAE/100 g FW) was determined in *G. ciliata*, while the highest (1365.92 mg GAE/100 g FW), with an almost 10 times higher value, in *E. annuus*. The species *R. japonica* and *S. gigantea* also showed high TPC values, on average, 921.71 mg GAE/100 g FW. TNFC values in the different IAS species ranged from 42.74 mg GAE/100 g FW (*G. ciliata*) to 878.92 mg GAE/100 g FW (*E. annus*). Of the IAS species analyzed, in addition to the aforementioned *E. annus*, the species *R. japonica* and *S. halepense* also had high TNFC contents. The TFC values ranged from 102.89 mg GAE/100 g FW (*G. ciliata*) to 647.5 mg GAE/100 g FW (*R. japonica*), while, for example, the species *E. annus* and *S. halepense* also showed high TFC values, on average, 497 mg GAE/100 g FW, and the species *A. theophrasti*, *A. retroflexus*, *A. artemisiifolia*, and *D. stramonium* showed the lowest TFC values, on average, 153 mg GAE/100 g FW. All total phenolic compounds analyzed in the IAS showed high values, suggesting that these species represent a highly valuable nutritional source and could be a potential source of phenolic, flavonoid, and nonflavonoid compounds for further use. Our results show significant differences in polyphenolic content among species, from which we can conclude that phenolic content depends on the habitat and environmental factors of a given species and is also strongly influenced by the genetic characteristics of species. The invasion success of terrestrial plant species, mostly IAS, has been linked to the environment-dependent production of specialized metabolites. When plant species are exposed to more biotic and abiotic stresses from the environment, the synthesis and accumulation of specialized metabolites are greater. As part of their defense mechanism, plants induce an immune system and tend to synthesize antioxidants (especially phenols) to detoxify the accumulated ROS. The IAS exhibit different defense mechanisms against biotic, abiotic, and anthropogenic factors, which is the reason why they occupy larger areas than other plant species, produce larger amounts of specialized metabolites, especially flavonoids, and thus create stronger resistance to environmental conditions [47,48]. Other authors [49] also reported high values of TPC in some IAS (*S. halepense*), as a potential source of phenolic and antioxidant compounds [50].

### 2.2. Pigment Compounds Content

Pigment compounds such as carotenoids, anthocyanins, betalains, and chlorophylls play an important role in plant organisms, both in primary and secondary metabolism and, above all, are strong antioxidants thus significantly contribute to the removal of the free radicals [51,52,53]. The results of pigment compounds analyzed in selected IAS, namely, chlorophyll a (TCh-a), chlorophyll b (TCh-b), total chlorophyll (TCh), and total carotenoid content (TCA), are shown in Table 2. According to the statistical analysis, highly significant differences (*p* ≤ 0.0001) were determined among the studied IAS. The TCh-a values ranged from 0.58 mg/g (*A. theophrasti*) to 1.05 mg/g (*G. ciliate*), while TCh-b from 0.35 mg/g (*A. retroflexus*) to 0.92 mg/g (*S. gigantea*). Based on the individual chlorophyll pigments content, the lowest total chlorophyll (TCh) content was determined for species (*A. theophrasti*), 1 mg/g, while the highest was for species *G. ciliate*, 1.81 mg/g. With the species *G. ciliate*, the highest TCh value was also obtained for the species *S. gigantea*, and according to the statistical analysis, they did not differ significantly. Additionally, species *A. retroflexus* and *A. artemisiifolia* did not differ significantly in TCh, with an average value of 1.215 mg/g, while also species *D. stramonium* and *E. annuus* had identical TCh values, 1.21 mg/g. TCA values ranged from 0.07 mg/g (*D. stramonium*) to 0.26 mg/g (*A. retroflexus*), and if we exclude those IAS with the highest and lowest values, we can observe that almost all remaining species (*A. artemisiifolia*, *E. annuus*, *G. ciliata*, *R. japonica*, *S. gigantea*, and *S. halepense*) did not differ statistically in TCA content, with an average value of 0.21 mg/g. Light exposure (UV radiation) is one of the key factors affecting pigment synthesis in the plant leaf. Therefore, in general, the content and accumulation of plant pigment strongly depend on irradiation. When the plants are exposed to too much UV radiation, they activate their photoprotective role and defense mechanisms and reduce both stomatal conduction and activity of RUBISCO enzymes, leading to reduced chlorophyll synthesis and thus the reduced rate of photosystem I and II [54]. Referring to the pedoclimatic conditions of the area where IAS within this study were collected (Section 3.1), a relatively low number of sunny days can be observed (which is a characteristic of the mentioned climate), thus explaining the relatively high content of pigment compounds in all IAS species analyzed. In general, all IAS studied contain significant content of pigment compounds, mainly chlorophylls and carotenoids. Based on the obtained results of pigment compounds, IAS can be considered as strong antioxidants and thus possess strong antimutagenic properties but also can find application as natural coloring agents (plant source) in the dye industry [53].

### 2.3. Antioxidant Capacity (ABTS)

Natural antioxidants typically include compounds (e.g., polyphenols, carotenoids, tocopherols, and ascorbic acid) that have both nutritional (reducing oxidative stress, preventing cancer, atherosclerosis, aging) and functional properties, as well as important health (e.g., reducing degenerative diseases, cancers, etc.) or industrial (unique flavors, additives) benefits [55].

The determined values of antioxidant capacity (ABTS) of analyzed IAS were high (Figure 3), ranging from 1666.96 (*S. halepense*) to 2221.97 µmol TE/kg (*A. retroflexus*), indicating that the analyzed plant material is of great importance from a functional point of view. Plants have efficient complex enzymatic and nonenzymatic antioxidant defense systems to avoid the toxic effects of free radicals, and since IAS are often exposed to stressful environments, anthropogenic effects, and are easily adaptable to the rapid spread and strongly competitive [11,14,19,20], higher amounts of ABTS are expected. The obtained data show that species *A. retroflexus*, *A. artemisiifolia*, *E. annuus*, and *R. japonica* did not differ significantly, with the average ABTS value of 2178.63 µmol TE/kg, indicating that the obtained results of this research are high for almost all IAS studied (Figure 3). It is worth mentioning that, as expected, the correlations and the first-order polynomial (FOP) model for ABTS calculation showed that when the *AsA* (*R. japonica*), TPC (*E.annuus*), or TCA (*A.retroflexus*) values are higher, the ABTS value is also higher. As mentioned, plants, particularly IAS, have an innate ability to synthesize nonenzymatic antioxidants—first, to synthesize a variety of phytochemicals to perform their normal physiological functions and protection from microbial pathogens and animal herbivores, and second, to respond to environmental stress conditions [56]. Under exposure to biotic and abiotic stressors, according to which IAS are very often exposed, the production of ROS in plant cells increases, leading to the induction of oxidative stress, which results in the accumulation of antioxidants, thus increasing the antioxidant capacity of plant sample [57,58].

### 2.4. Correlation Matrix between the Observed Parameters

The positive correlation (Table 3) between TPC and TNFC content was observed, statistically significant at *p* < 0.001 level, while the correlation between TPC and TFC was positive, statistically significant at *p* < 0.05 level. A positive but statistically nonsignificant correlation (*p* > 0.05) was observed between TFC and *AsA* content. The content of TCh was positively correlated to TCh-a and TCh-b content and statistically significant at *p* < 0.05 level. ABTS value was positively correlated to TCA content and statistically significant at *p* < 0.05 level. According to performed analysis, it can be concluded that correlations between IAs species with increased TPC content would lead to higher TFC and TNFC values. The correlation analysis confirmed that the higher TCA values contributed to the augmented ABTS values, or that IAs species with the increased TCA content exerted a more pronounced antioxidant capacity.

### 2.5. Cluster Analysis

A dendrogram of samples according to their bioactive compounds was calculated using complete linkage as an amalgamation rule and the city block (Manhattan) distance, which was used as a measure of the proximity between samples, was presented in Figure 4. The dendrogram based on obtained data showed a proper distinction between samples. According to the bioactive compounds obtained in the samples, two clusters were identified. Samples of *R. japonica*, *S. gigantea* and *E. annuus* were grouped in the first cluster, which was categorized as the group with the highest values of TFC, TPC, TNFC, and DM. The second cluster covered samples of *A. artemisifolia*, *G. ciliata* and *A. retroflexus*, *A. theophrasti*, *D. stramonium*, and *S. halepense*, which were outlined as plant samples with lower TFC, TPC, TNFC, and DM values.

### 2.6. Principal Component and First-Order Polynomial Model

The points presented in the PCA graphics, which were in close vicinity to each other (in the geometric sense) indicate the possible closeness of variables that represent these points. The direction of the vector defining the variable in factor space announces an augmenting trend of the variable, while the magnitude of the vector corresponds to the square of the correlation. The angles between vectors presenting two variables in factor space indicate the degree of their correlations (smaller angles are assigned to higher correlations).

The PCA of the presented bioactive compounds data explained that the first three principal components covered 78.05% of the total variance (these components explained 30.82; 27.67 and 19.56%, respectively, while the obtained eigenvalues were 3.08, 2.77, and 1.96, respectively) in 10 variables factor space (bioactive compounds contents). According to the PCA graphic, DM (which contributed 9.2% of the total variance, based on correlations), the content of TNFC (24.9%), TPC (29.6%), and TFC (23.4%) exhibited negative scores to the first principal component calculation (Figure 5). The positive contribution to the second principal component calculation was observed for the content of TCh-a (34.3% of the total variance, based on correlations), TCh-b (9.2%), TCA (15.7%), and TCh (25.9%). The positive contribution to the third principal component was obtained for ABTS (34.3% of the total variance, based on correlations) and the content of TCA (21.3%), while negative influence was observed for the content of TCh-b (24.5%) and TCh (11.1%).

The map of PCA showed that the PC1 coordinate described the differentiation among the variables TFC, TPC, TNFC, and DM, while the PC2 coordinate explained the variations in TCh-a, TCh-b, TCc, and TCA between samples. The PC3 coordinate showed the versatility of ABTS between samples. According to the PCA graphic, the most intensive differentiation was observed between two groups of samples: (1) *R. japonica*, *S. gigantea* and *E. annuu**s* (which are categorized by high TFC, TPC, TNFC, and DM values, located at the left side of the PCA graphic) and (2) the group of samples located at the right side of the graph (*A. artemisifolia*, *G. ciliata*, *A. retroflexus*, *A. theophrasti*, *D. stramonium*, and *S. halepense*, with minor TFC, TPC, TNFC, and DM values). The sample of *G. ciliata* showed the highest TCh-a value, while *E. annuus*, *R. japonica*, *A. retroflexus*, and *A. artemisifolia* showed increased values of ABTS and TCA.

The ANOVA analysis was conducted for the developed FOP model for ABTS calculation, and the response variable was tested against the impact of independent variables (Table 4). According to ANOVA results, ABTS content was mostly affected by TCA, statistically significant at *p* < 0.05 level, while TFC was also very influential, at *p* < 0.05 level.

The FOP model for ABTS calculation could be presented according to the following equation:(1)$$\begin{array}{cc} \mathrm{ABTS} & =1192.212+17.287\cdot \mathrm{AsA}+0.862\cdot \mathrm{TPC}\;-\;0.358\cdot \mathrm{TNFC}\\ & -\;1.517\cdot \mathrm{TFC}\;-\;80.678\cdot \mathrm{TCh}+3228\cdot \mathrm{TCA} \end{array}$$

The developed first-order polynomial (FOP) model illustrated a statistically insignificant shift from the experimental data, which corroborated its suitability. The evaluated values of r^2^ (0.990), P (0.736), RMSE (20.457), and χ^2^ (470.814) confirmed that obtained FOP model was statistically significant and in agreement with experimental results. The residual variance, marked as “Error” in Table 4, explained the model discrepancy with the obtained ABTS values, i.e., the participation of terms that were not described in the FOP model. The residual analysis indicated that the mean of residuals was zero, while the standard deviation was equal to 21.698 (with a minimum of −26.335 and a maximum of 48.323). The skewness parameter (1.393) expressed a minimal departure of residuals from the normal distribution, while the kurtosis (2.777) demonstrated nearly negligible divergence in “peakedness,” compared to normal distribution.

## 3. Materials and Methods

### 3.1. Plant Material

Invasive plant species *Abutilon theophrasti*, *Amaranthus retroflexus*, *Ambrosia artemisiifolia*, *Datura stramonium*, *Erigeron annuus*, *Galinsoga ciliata*, *Reynoutria japonica*, *Solidago gigantea*, and *Sorghum halepense* were collected in the Žumberak mountain massif in Croatia. The study was conducted in the period from April to June 2019, depending on the flowering stage of IAS. Specifically, all selected IAS in the Žumberak area were collected just before the flowering stage for each species as follows: *D. stramonium*, *E. annuus*, *G. ciliata* and *S. halepense* in April; *A. theophrasti*, *A. artemisiifolia* and *R. japonica* in May; and *A. retroflexus* and *S. gigantea* in June (Table 5). The detailed inventory, botanical determination, and classification of IASs were based on various literature sources [13,59,60,61]. In addition, meteorological data, and soil type of the aforementioned area as well as the exact locations of where the species were collected and are presented in Table 5. Žumberak is a specific area rich in native flora, with increasing occurrence of invasive plant species that threaten the biodiversity of the protected area. From each invasive plant species, approximately 2 kg of fresh weight biomass was collected per sample in three replicates. All plant collections were carried out in the early morning hours in dry weather. Only healthy above-ground parts of plants (stems and leaves without roots) were collected and immediately after collection transported in paper bags to the laboratory of the Department of Agricultural Technology, Storage, and Transport at the University of Zagreb, Faculty of Agriculture, for further analysis. Samples were stored at 4 °C for up to 3 days till the intended chemical analysis.

### 3.2. Determination of Dry Matter Content, Bioactive Compounds Content, and Antioxidant Capacity

The analysis of dry matter content (%) was conducted by drying fresh IAS biomass (leaf part) till constant mass, according to the standard method [64], at 105 °C.

Ash content was determined according to the standard method [65]. Fresh IAS biomass was first dried at 105 °C to a constant dry matter content. Then, 1 g of the dry biomass sample was weighed into the crucible (in three replicates) and placed in the cold muffle furnace (Nabertherm L-401K2TN, Lilienthal, Germany), where after even raise of temperature to 550 °C, further heating was carried out at 550 °C for at least 120 min more. Simultaneous determination of total carbon, hydrogen, sulfur, and nitrogen was carried out by combustion in CHNS analyzer (Elementar, Vario Macro) using helium (He) as carrier gas. The 50 mg of dry sample was wrapped in aluminum foil and placed in the carousel (autosampler) along with tungsten oxide powder (WO_3_) as the catalyst, prepared in three replicates and placed into the carousel (autosampler), after which was combusted in the furnace of the instrument according to the standard method [66] for total C, H, and N content, and for total S content [67]. The relative percentage of total oxygen (O) was calculated by subtracting the relative percentage of total C, H, N, and S from 100. The amount of protein was then determined by multiplying the nitrogen content by a factor of 6.25, as indicated in various relevant studies in the literature [68,69].

From bioactive compounds, the following were analyzed: vitamin C (*AsA*) content (mg/100 g FW) by titration with 2,6-dichlorindophenol (DKF) according to [70]. For each IAS species, 5 g ± 0.01 of fresh leaves were homogenized with 100 mL of 2% (*v*/*v*) oxalic acid. After homogenization, the solution was filtered through Whatman filter paper, the filtrate was obtained in a volume of 10 mL and titrated with freshly prepared 2,6-dichlorindophenol until the appearance of pink coloration of tested sample solution. The final *AsA* content was calculated according to equation (2) and expressed as mg/100 g fresh weight.
(2)$$AsA=\frac{V\;\left(DKF\right)\times F}{D}\times 100$$
where *V* (*DKF*)—volume of *DKF* (mL); *F*—factor of DKF; *D*—sample mass used for titration.

Total phenolics (TPC), flavonoids (TFC), and nonflavonoids (TNFC) content were determined spectrophotometrically (Shimadzu, UV 1900i, Duisburg, Germany) based on the method described by Ough and Amerine [71]. The method is based on a color reaction that phenolics develop with the Folin–Ciocalteu reagent, which is measured spectrophotometrically at 750 nm. For the purpose of polyphenolic compound extraction from plant leaves, ethanol solution (80%, *v*/*v*) and reflux were combined. Fresh plant leaves of 10 g ± 0.01 were weighed into an Erlenmeyer flask, and the first 40 mL of 80% EtOH (*v*/*v*) was added. The prepared sample was heated to boiling point and additionally refluxed for 10 min. After 10 min, the sample was filtered through Whatman filter paper into a volumetric flask of 100 mL. After filtration, the rest of the sample was transferred to the Erlenmeyer flask, and another 50 mL of 80% EtOH (*v*/*v*) was added while the procedure was repeated with reflux for 10 min. The sample was filtered, the filtrates were combined, and the flask was made up to the mark with 80% EtOH (*v*/*v*). The extract thus prepared was used for the reaction with the Folin–Ciocalteu reagent according to the following procedure: to a volumetric flask of 50 mL, 0.5 mL of the plant ethanolic extract was added, then 30 mL of distilled water (dH_2_O), 2.5 mL of the prepared Folin–Ciocalteu reagent (1:2 with dH_2_O), and 7.5 mL of saturated sodium carbonate solution (Na_2_CO_3_). The flask was made up to the mark with dH_2_O and the reaction was allowed to stand at room temperature for 2 h with intermittent shaking. The same ethanolic extracts prepared for TPC were used for TNFC content determination. TNFC separation was performed according to the following procedure: a total of 10 mL of the ethanolic extract was added to the 25 mL capacity volumetric flask, and then 5 mL of HCl (1:4, *v*/*v*) and 5 mL of formaldehyde were added. The prepared samples were treated with nitrogen (N_2_) and left for 24 h at room temperature in a dark place. After 24 h, the same Folin–Ciocalteu reaction was performed as for TPC. The absorbance of blue color in both TPC and TNFC reactions was measured spectrophotometrically at 750 nm using distilled water as a blank. Gallic acid and catechol were used as external standards, and the concentration of TPC and TNFC content was expressed as mg GAE/100 g fresh weight. TFC content was mathematically expressed as the difference between total phenols and nonflavonoids.

Analyses of pigment compounds included the determination of total chlorophylls and carotenoids according to the method described by Holm [72] and Wettstein [73]. For extraction purposes of pigments, 0.2 g ± 0.01 of fresh plant leaves were weighed, and a total volume of 15 mL of acetone (p.a.) was added three times. After each addition of acetone, the samples were homogenized using a laboratory homogenizer (IKA, UltraTurax T-18, Staufencity, Germany). The final solution was filtered and transferred to a volumetric flask of 25 mL. The absorbance was measured spectrophotometrically (Shimadzu UV 1900i, Duisburg, Germany) at 662, 644, and 440 nm using acetone as a blank. To obtain the results for pigment content based on the measured absorbance, the equations of Holm–Wettstein were used (3). The final results for the pigments were expressed in mg/g.
(3)$$\begin{array}{c} chlorophyll\;a=9.784\times A_{662}-0.990\times A_{644}\;\left[mg/L\right]\\ chlorophyll\;b=21.426\times A_{644}-4.65\times A_{662}\;\left[mg/L\right]\\ total\;chlorophyll=5.134\times A_{662}+20.436\times A_{644}\;\left[mg/L\right]\\ total\;carotenoids=4.695\times A_{440}-0.268\times total\;chlorophyll\;\left[mg/L\right] \end{array}$$

For the determination of antioxidant capacity, ABTS assay was performed according to Miller et al. [74]. ABTS, 2,2′-azinobis (3-ethylbenzothiazoline-6-sulfonic acid), and potassium persulfate were obtained from Sigma-Aldrich. As antioxidant standard Trolox (6-hydroxy-2,5,7,8-tetramethylchroman-2-carboxylic acid; Sigma-Aldrich, St. Louis, MO, USA) was used, and a stock standard Trolox (2.5 mM) was prepared in ethanol (80% *v*/*v*). To prepare ABTS radical solution (ABTS•+) the 5 mL of ABTS solution (7 mM) and 88 mL of potassium persulfate (140 mM) solution were mixed and allowed to stand in the dark at room temperature for 16 h. On the day of analysis, 1% ABTS•+ solution (in 96% ethanol) was prepared. A total of 160 µL of ethanolic extract were directly injected into the cuvette and mixed with 2 mL 1% ABTS•+, while absorbance at 734 nm was measured (Shimadzu 1900i, Duisburg, Germany). The final results of the antioxidant capacity were calculated based on a calibration curve and expressed as µmol TE/L.

### 3.3. Statistical Analysis

All measurements were done in triplicate, data was averaged, expressed by means ± standard deviation (SD). The ANOVA and Duncan’s multiple range tests at 99% confidence limit were performed to show the variations in means between samples using statistical software SAS [75] (mean values were compared by an LSD test where *p* = 1% was considered the statistical level of significance). Values are presented as means of three determinations ± standard deviation. Values in the same column with the different superscript lowercase letters (a–e) indicate significant differences between means at *p* ≤ 0.0001. The average deviation of the results from the mean value for each investigated parameter with the values of standard deviation is also presented.

Principal component analysis (PCA) was used to investigate the possible correlations between the measured variables, while cluster analysis (CA) was applied to classify objects into groups. The city-block (Manhattan) distance metrics were used, and the complete linkage method was utilized for amalgamation calculation in the CA analysis.

The first-order polynomial (FOP) model was used to estimate the main effect of the variables on ABTS. The independent variables used for modeling were the contents of *AsA*, TPC, TNFC, TFC, TCh, and TCA, while ABTS was the response variable. The FOP model was fitted to data collected from experimental measurements as follows:(4)$$Y=\beta_{0}+\sum_{i=1}^{6}\beta_{i}X_{i}$$
where *β*_0_ and *β_i_* were constant regression coefficients, and *Y* was the response variable, while *X_i_* were independent variables. The significant terms in the model were discovered using ANOVA for each dependent variable.

The goodness of fit for the developed model was evaluated using the coefficient of determination (r^2^), the mean relative percent error (P), the root mean square error (RMSE), and the reduced chi-square (χ^2^). The higher the values of r^2^ and the lower the values of P, RMSE, and χ^2^ are, the better is the goodness of the fit. These parameters can be calculated as follows:(5)$$P=\frac{100}{N}\cdot \sum_{i=1}^{n}\frac{\left|Y_{\exp,i}-Y_{pre,i}\right|}{Y_{\exp,i}},\;RMSE={\left[\frac{1}{N}\sum_{i=1}^{n}{\left(Y_{\exp,i}-Y_{pre,i}\right)}^{2}\right]}^{1/2},\;\chi^{2}=\frac{\sum_{i=1}^{n}{\left(Y_{\exp,i}-Y_{pre,i}\right)}^{2}}{N-n}$$
where *Y_exp_*_,*i*_ is the ith experimentally observed response *Y*, *Y_pre_*_,*i*_ is the ith predicted *Y*, and N is the number of observations and n is the number of constants.

The data were studied by applying the STATISTICA 10.0 software [76].

## 4. Conclusions

Based on the obtained results, it can be concluded that the analyzed IAS have significantly high amounts of bioactive compounds as well as favorable chemical and nutritional composition for further utilization. The highest value of ash was obtained by *G. ciliata*, while *R. japonica* had the highest values of protein and total nitrogen content. *S. gigantea* had the highest value of total carbon and total hydrogen and also the lowest values of sulfur. *R. japonica* was notable with high ascorbic acid content and total flavonoid content, while *E. annuus* had the highest total phenol content. The highest values of total chlorophyll content were obtained by *G. ciliata*, while *A. retroflexus* had the highest total carotenoid content. The most prominent results of antioxidant capacity were obtained by *A. retroflexus.* Based on these results, it can be concluded that investigated IAS have high nutrient potential and can be considered as valuable plant material, respectively, a significant source of natural antioxidants for further utilization. Selected plants with high ABTS values (*A. retroflexus*, *A. artemisiifolia*, *E. annuus*, and *R. japonica*) could be interesting from the point of view of further use, e.g., in the production and development of pharmaceuticals, phytotherapy, additives, biopesticides, herbicides, etc. Furthermore, this research suggests that IAS can be viewed in a new light, not as a troublesome nuisance but as a potentially sustainable source of bioactive compounds with high nutritional value and beneficial antioxidants. On the other hand, large-scale harvesting and utilization of aerial plant parts (zero waste) of IAS is encouraged (especially in protected areas where they pose a threat to biodiversity) and may represent a powerful and economically driven method of mechanical control of these highly invasive and widespread alien species. Finally, it should be noted that further research should focus on more detailed analytical studies (HPLC methods) of individual phytochemicals, especially phenolic compounds, to identify specific phytonutrients and their further potential for use in specific process branches.

## Figures and Tables

**Figure 1 molecules-26-03814-f001:**
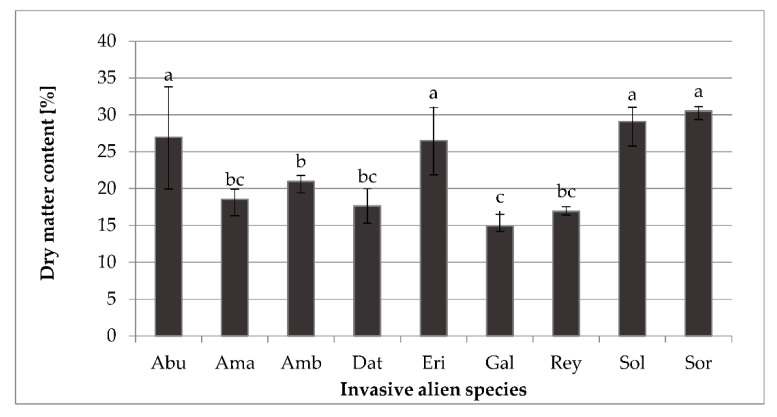
Dry matter content (%) of nine different invasive alien species. Abu—*Abutilon theophrasti*; Ama—*Amaranthus retroflexus*; Amb—*Ambrosia artemisiifolia*; Dat—*Datura stramonium*; Eri—*Erigeron annuus*; Gal—*Galinsoga ciliata*; Rey—*Reynoutria japonica*; Sol—*Solidago gigantea*; Sor—*Sorghum halepense*. Different letters (a–c) indicate significant differences between means at *p* ≤ 0.0001.

**Figure 2 molecules-26-03814-f002:**
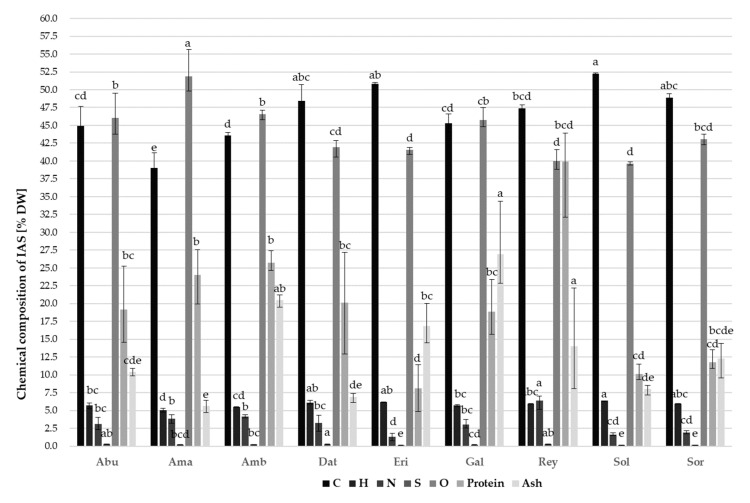
Chemical composition of different IAS. DW—dry weight; Abu—*Abutilon theophrasti*; Ama—*Amaranthus retroflexus*; Amb—*Ambrosia artemisiifolia*; Dat—*Datura stramonium*; Eri—*Erigeron annuus*; Gal—*Galinsoga ciliata*; Rey—*Reynoutria japonica*; Sol—*Solidago gigantea*; Sor—*Sorghum halepense*. Data represent the mean of three analytical measurements ± SD. Different letters (a–e) indicate significant differences between means at *p* ≤ 0.0001.

**Figure 3 molecules-26-03814-f003:**
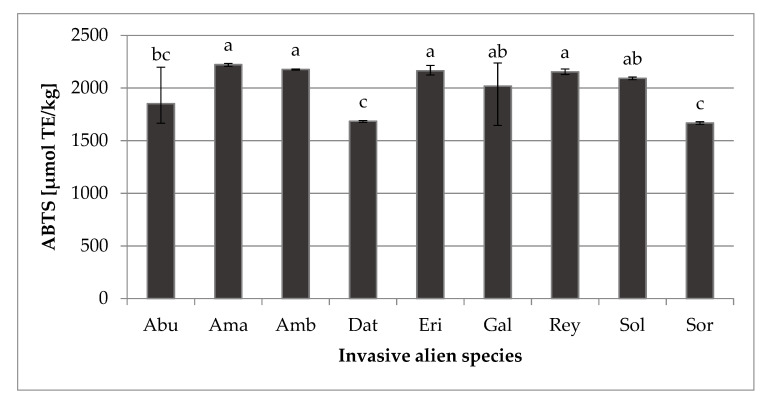
ABTS—antioxidant capacity (µmol TE/kg) of nine different invasive alien species. Abu—*Abutilon theophrasti*; Ama—*Amaranthus retroflexus*; Amb—*Ambrosia artemisiifolia*; Dat—*Datura stramonium*; Eri—*Erigeron annuus*; Gal—*Galinsoga ciliata*; Rey—*Reynoutria japonica*; Sol—*Solidago gigantea*; Sor—*Sorghum halepense*. Different letters (a–c) indicate significant differences between means at *p* ≤ 0.0001.

**Figure 4 molecules-26-03814-f004:**
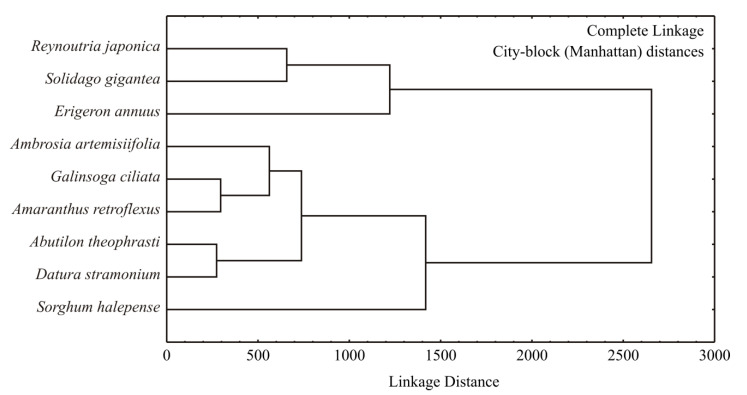
Complete linkage dendrogram for specialized metabolites of IAS.

**Figure 5 molecules-26-03814-f005:**
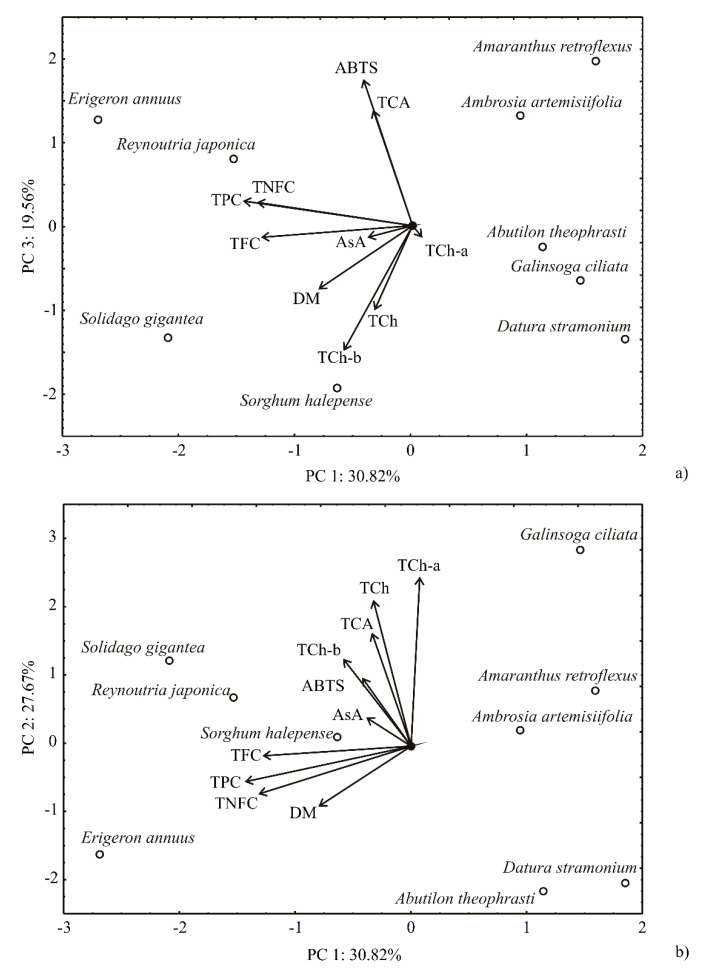
PCA ordination of variables based on component correlations: (**a**) projection in PC1–PC3 factor plane; (**b**) projection in PC1–PC2 factor plane. DM—dry matter; TPC- total phenol content; TNFC—total nonflavonoid content; TFC—total flavonoid content; TCh—total chlorophyll content; TCh-a—chlorophyll a content; TCh-b—chlorophyll b content; TCA—total carotenoid content; ABTS—antioxidant capacity; *AsA*—vitamin C content.

**Table 1 molecules-26-03814-t001:** Bioactive compounds content obtained from invasive plant species biomass.

Sample	*AsA* [mg/100 g FW]	TPC [mg GAE/100 g FW]	TNFC [mg GAE/100 g FW]	TFC [mg GAE/100 g FW]
*A. theophrasti*	20.50 ± 4.42 ^bc^	334.03 ± 80.66 ^d^	157.18 ± 47.65 ^cde^	176.86 ± 35.10 ^d^
*A. retroflexus*	19.51 ± 0.39 ^bc^	187.15 ± 62.59 ^de^	82.02 ± 33.26 ^de^	105.12 ± 29.96 ^d^
*A. artemisiifolia*	19.64 ± 0.91 ^bc^	342.21 ± 75.99 ^d^	162.20 ± 37.94 ^cde^	180.00 ± 38.27 ^d^
*D. stramonium*	21.36 ± 3.29 ^bc^	285.62 ± 73.91 ^de^	134.99 ± 37.84 ^de^	150.64 ± 36.41 ^d^
*E. annuus*	16.00 ± 3.61 ^c^	1365.92 ± 185.74 ^a^	878.92 ± 203.57 ^a^	487.00 ± 24.61 ^b^
*G. ciliata*	19.13 ± 5.68 ^bc^	142.66 ± 40.79 ^e^	42.74 ± 37.57 ^e^	102.89 ± 40.39 ^d^
*R. japonica*	38.46 ± 9.39 ^a^	937.72 ± 75.39 ^b^	290.11 ± 21.89 ^c^	647.50 ± 64.44 ^a^
*S. gigantea*	21.66 ± 4.84 ^bc^	905.69 ± 95.87 ^b^	541.56 ± 80.37 ^b^	364.13 ± 26.12 ^c^
*S. halepense*	25.08 ± 4.50 ^b^	506.94 ± 88.83 ^c^	205.36 ± 59.59 ^cd^	506.94 ± 88.83 ^b^
ANOVA	*p* ≤ 0.0013	*p* ≤ 0.0001	*p* ≤ 0.0001	*p* ≤ 0.0001

Values are presented as means of three determinations ± standard deviation. Values in the same column with the different superscript lowercase letters (a–e) indicate significant differences between means at *p* ≤ 0.0001; *AsA*—ascorbic acid content; TPC—total phenol content; TNFC—total nonflavonoid content; TFC—total flavonoid content.

**Table 2 molecules-26-03814-t002:** Pigment compounds content obtained from nine different invasive alien species.

Sample	TCh-a [mg/g]	TCh-b [mg/g]	TCh [mg/g]	TCA [mg/g]
*A. theophrasti*	0.58 ± 0.21 ^d^	0.42 ± 0.11 ^d^	1.00 ± 0.12 ^d^	0.14 ± 0.13 ^bc^
*A. retroflexus*	0.87 ± 0.07 ^ab^	0.35 ± 0.07 ^d^	1.21 ± 0.13 ^bcd^	0.26 ± 0.02 ^a^
*A. artemisiifolia*	0.80 ± 0.16 ^bc^	0.42 ± 0.17 ^d^	1.22 ± 0.32 ^bcd^	0.23 ± 0.03 ^ab^
*D. stramonium*	0.58 ± 0.05 ^d^	0.54 ± 0.04 ^bcd^	1.12 ± 0.09 ^cd^	0.07 ± 0.01 ^c^
*E. annuus*	0.63 ± 0.07 ^cd^	0.50 ± 0.06 ^cd^	1.13 ± 0.13 ^cd^	0.21 ± 0.04 ^ab^
*G. ciliata*	1.05 ± 0.04 ^a^	0.76 ± 0.20 ^ab^	1.81 ± 0.21 ^a^	0.22 ± 0.06 ^ab^
*R. japonica*	0.82 ± 0.16 ^bc^	0.53 ± 0.08 ^cd^	1.35 ± 0.23 ^bc^	0.21 ± 0.05 ^ab^
*S. gigantea*	0.85 ± 0.07 ^b^	0.92 ± 0.04 ^a^	1.77 ± 0.04 ^a^	0.19 ± 0.04 ^ab^
*S. halepense*	0.87 ± 0.07 ^ab^	0.65 ± 0.20 ^bc^	1.52 ± 0.27 ^ab^	0.18 ± 0.05 ^ab^
ANOVA	*p* ≤ 0.0010	*p* ≤ 0.0005	*p* ≤ 0.0004	*p* ≤ 0.0290

Values are presented as means of three determinations ± standard deviation. Values in the same column with the different superscript lowercase letters (a–d) indicate significant differences between means at *p* ≤ 0.0001; TCh-a—chlorophyll a; TCh-b—chlorophyll b; TCh—total chlorophyll content; TCA—total carotenoid content.

**Table 3 molecules-26-03814-t003:** Correlation matrix between the observed parameters.

	*AsA*	TPC	TNFC	TFC	TCh-a	TCh-b	TCh	TCA	ABTS
DM	−0.204	0.422	0.508	0.369	−0.265	0.239	0.005	−0.130	−0.266
*AsA*		0.196	−0.144	0.645 **	0.144	0.054	0.113	0.012	−0.012
TPC			0.941 ^++^	0.797 *	−0.265	0.209	−0.014	0.125	0.320
TNFC				0.578	−0.326	0.217	−0.042	0.097	0.310
TFC					−0.040	0.201	0.104	0.101	0.052
TCh-a						0.458	0.832 ^+^	0.645	0.274
TCh-b							0.875 ^+^	−0.078	−0.165
TCh								0.303	0.047
TCA									0.821 ^+^

Correlation statistically significant at: *++*
*p* < 0.001 level; *+*
*p* < 0.01 level; * *p* < 0.05 level; ** *p* < 0.10 level. DM—dry matter; TPC—total phenol content; TNFC—total nonflavonoid content; TFC—total flavonoid content; TCh—total chlorophyll content; TCh-a—chlorophyll a content; TCh-b—chlorophyll b content; TCA—total carotenoid content; ABTS—antioxidant capacity; *AsA*—vitamin C content.

**Table 4 molecules-26-03814-t004:** Analysis of variance (ANOVA) for FOP model for ABTS calculation.

Variable	dF	ABTS	*p*-Value
*AsA*	1	1597.8	0.454
TPC	1	1593.6	0.455
TNFC	1	125.5	0.82
TFC	1	73,841.0 *	0.025
TCh	1	2938.8	0.338
TCA	1	144,906.0 *	0.013
Error	2	3766.5	

* Significant at *p* < 0.05 level, dF—degrees of freedom.

**Table 5 molecules-26-03814-t005:** Geographical coordinates, meteorological data [62], and soil type [63] of collected IAS on the Žumberak area.

Invasive Alien Plant Species			
*A. theophrasti*			
Coordinates	45° 41′ 33.7812″ N 15° 29′ 24.9684″ E	45° 39′ 46.3716″ N 15° 35′ 26.8476″ E	45° 40′ 51.4488″ N 15° 30′ 49.4856″ E
Dominant Soil type	Rendzina on marl and soft limestone	Pseudogley on sloping terrain	Rendzina on marl and soft limestone
Time of sampling	May	May	May
*A. retroflexus*			
Coordinates	45° 41′ 33.7812″ N 15° 29′ 24.9684″ E	45° 44′ 57.9120″ N 15° 23′ 30.6384″ E	45° 47′ 26.2824″ N 15° 29′ 29.8608″ E
Dominant Soil type	Rendzina on marl and soft limestone	Rendzina on marl and soft limestone	Cambisol eutric on flysch or soft limestone
Time of sampling	June	June	June
*A. artemisiifolia*			
Coordinates	45° 41′ 38.9364″ N 15° 29′ 15.6480″ E	45° 45′ 10.8360″ N 15° 25′ 25.9284″ E	45° 44′ 18.9564″ N 15° 28′ 14.2284″ E
Dominant Soil type	Rendzina on marl and soft limestone	Rendzina on marl and soft limestone	Cambisol eutric on flysch or soft limestone
Time of sampling	May	May	May
*D. stramonium*			
Coordinates	45° 39′ 46.3716″ N 15° 35′ 26.8476″ E	45° 40′ 49.9620″ N 15° 30′ 39.2364″ E	45° 40′ 8.9148″ N 15° 34′ 58.2960″ E
Dominant Soil type	Pseudogley on sloping terrain	Rendzina on marl and soft limestone	Pseudogley on sloping terrain
Time of sampling	April	April	April
*E. annuus*			
Coordinates	45° 39′ 25.2108″ N 15° 31′ 14.0376″ E	45° 40′ 45.2532″ N 15° 30′ 47.2896″ E	45° 40′ 49.8972″ N 15° 30′ 42.5268″ E
Dominant Soil type	Calcocambisol on dolomite	Pseudogley on sloping terrain	Rendzina on marl and soft limestone
Time of sampling	April	April	April
*G. ciliata*			
Coordinates	45° 41′ 33.7812″ N 15° 29′ 24.9684″ E	45° 43′ 16.7520″ N 15° 26′ 12.5808″ E	45° 40′ 49.9620″ N 15° 30′ 39.2364″ E
Dominant Soil type	Rendzina on marl and soft limestone	Rendzina on skeletal limestone	Rendzina on marl and soft limestone
Time of sampling	April	April	April
*R. japonica*			
Coordinates	45° 50′ 3.8400″ N 15° 37′ 3.5616″ E	45° 50′ 5.7660″ N 15° 36′ 52.3008″ E	45° 46′ 08.7492” N 15° 24′ 48.9996” E
Dominant Soil type	Rendzina, Luvisol, Cambisol on Terra rossa	Rendzina, Luvisol, Cambisol on Terra rossa	Cambisol eutric on flysch or soft limestone
Time of sampling	May	May	May
*S. gigantea*			
Coordinates	45° 40′ 49.8972″ N 15° 30′ 42.5268″ E	45° 50′ 21.8904″ N 15° 35′ 55.572″ E	45° 43′ 16.7520″ N 15° 26′ 12.5808″ E
Dominant Soil type	Rendzina on marl and soft limestone	Rendzina, Luvisol, Cambisol on Terra rossa	Rendzina on skeletal limestone
Time of sampling	June	June	June
*S. halepense*			
Coordinates	45° 39′ 46.3716″ N 15° 35′ 26.8476″ E	45° 40′ 51.4488″ N 15° 30′ 49.4856″ E	45° 40′ 8.9148″ N 15° 34′ 58.2960″ E
Dominant Soil type	Pseudogley on sloping terrain	Rendzina on marl and soft limestone	Pseudogley on sloping terrain
Time of sampling	April	April	April
Average air temperature (°C)	9.8	9.8	9.8
Average precipitation (mm)	1315.2	1315.2	1315.2
Relative air humidity (%)	71	71	71
Number of sunny days	56	56	56

## Data Availability

Data generated during the study can be obtained by the authors of this study.

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
