# Peer review of "Invasive Plant Species Biomass—Evaluation of Functional Value"

_molecules, 2021, doi:10.3390/molecules26133814_

Round 1
Reviewer 1 Report
In the submitted manuscript entitled “Recovering Bioactive Compounds from Invasive Plant Species Biomass”, the authors tried to evaluate the nutritional potential, phytochemical status, and antioxidant capacity of nine alien invasive plant species. The topic has some merit. However, the work is pretty preliminary in its current form. When I went through the manuscript, some points raised and needed to be fully addressed. Also, I feel this article outside the scope of this journal based on its contents. Rejection of the work is thus suggested.
Here are some major points.
- For most of the figures and tables, the positive control is missing, which make it hard to assess the major points the authors want to address. A thorough revision is terribly needed before it can be processed further.
- Since each and every invasive plant species may have its specific bioactive compounds, why the authors only select the four factors to compare and study such as AsA—Ascorbic acid content; TPC—total phenol content; TNFC— total non-flavonoid content; TFC— total flavonoid content? Why did the authors study total non-flavonoid content?
- The authors used bioactive compounds in the title, and actually they determined only the total contents of some very common phytochemical species, and even without a single compound identified. Thus, the title should be revised to reflect the study more accurately.
- Fingerprinting analysis of these species should at least be done, and more activity should be tested if the authors use the title in this work.
- The topic has some merit. However, I feel it may be outside the scope of this journal.
Author Response
Dear Reviewer,
we thank you for your valuable suggestions. We have noted all corrections and changes in the manuscript with the option "Track Changes" and highlighted by yellow colour the parts of the text that we have corrected according to your valuable suggestions. We have also highlighted the newly added references in the bibliography by green colour. Also, in the Letter from the attachment we in detail explained research topic.

Reviewer 2 Report
The aim of the study was to analyze selected Invasive alien species (IAS) in terms of their basic chemical composition, including the content of bioactive compounds and colorants. The weakest aspect of the manuscript and the planned experiment are the analytical methods chosen. All methods are based on titration or spectrophotometric measurements. A journal such as “Molecules” uses analytical techniques to determine the content of given compounds, i.e. chromatographic techniques. The choice of these methods is a major drawback of work. In the discussion of the results, a large part of the text concerns the introduction and properties of the compounds being determined. It is not needed in this chapter, where such introductory information should not be included. The strengths of the article are the applied statistical methods and the description of the introduction to the research, which convinces about the importance of the topic.
Detail comments:
Important information is missing in the description of the research material. How was the IAS species identified? In which year was the research conducted? At what stage were the plants tested? The samples were stored at 4 degrees Celsius until the analyzes were performed. How long it took?
L437 2,6-dichlorindophenol or 2,6-dichloroindophenol?
How was the color of the sample (2% oxalic acid extract) eliminated in determining the color change to slightly pink in determining the vitamin C content?
L496 2,20-azinobis (3-ethylbenzothiazoline-6-sulfonic acid) or 2,2′-azino-bis(3-ethylbenzothiazoline-6-sulfonic acid)
Why was such an ABTS radical cations (ABTS•1) notation used? The standard and most frequently used notation is ABTS•+
L514 - end the parenthesis
The chapter on Bioactive Compounds content provides a very extensive introduction to the topic. The beginning of this chapter resembles the introduction to the article, but not a discussion of the results obtained. I recommend that this introduction to the topic be shortened.
L189 - I think that the conclusion from the obtained results that IAS with vitamin C content at the level of 16-38 mg / 100 g is a material from which vitamin C should be extracted is exaggerated. Compared to other plant materials, these plants are low in vitamin C, so I would be careful to make such conclusions.
Table 4 - What is the difference in the description of table 4 in + p<0.01 level; and ** p<0.010 level?
Author Response

(The authors gave the same response as above.)

Round 2
Reviewer 1 Report
Since the authors have revised the manuscript to address most if not all of the points pointed out, and the revised manuscript is now improved. Thus, acceptance of the work is suggested.
Reviewer 2 Report
All my comments have been taken into account.